# MiRNA let-7 from TPO(+) Extracellular Vesicles is a Potential Marker for a Differential Diagnosis of Follicular Thyroid Nodules

**DOI:** 10.3390/cells9081917

**Published:** 2020-08-18

**Authors:** Lidia Zabegina, Inga Nazarova, Margarita Knyazeva, Nadezhda Nikiforova, Maria Slyusarenko, Sergey Titov, Dmitry Vasilyev, Ilya Sleptsov, Anastasia Malek

**Affiliations:** 1Subcellular Technology Lab., N.N. Petrov National Medical Research Center of Oncology, 195251 St. Petersburg, Russia; lidusikza@yandex.ru (L.Z.); oblaka12@mail.ru (I.N.); margo9793@gmail.com (M.K.); niki2naden_ka@mail.ru (N.N.); slusarenko_masha@mail.ru (M.S.); dvasilyev@hotmail.com (D.V.); 2Oncosystem Ltd., 121205 Moscow, Russia; 3Institute of Biomedical Systems and Biotechnologies, Peter the Great St. Petersburg Polytechnic University, 195251 St. Petersburg, Russia; 4PCR Laboratory; AO Vector-Best, 630117 Novosibirsk, Russia; titovse78@gmail.com; 5Department of endocrine surgery, Clinic of High Medical Technologies, St. Petersburg State University N.I. Pirogov, 190103 St. Petersburg, Russia; newsurgery@yandex.ru

**Keywords:** liquid biopsy, biomarker, thyroid cancer, follicular adenoma, EV, TPO, miRNA, Let-7

## Abstract

Background: The current approaches to distinguish follicular adenomas (FA) and follicular thyroid cancer (FTC) at the pre-operative stage have low predictive value. Liquid biopsy-based analysis of circulating extracellular vesicles (EVs) presents a promising diagnostic method. However, the extreme heterogeneity of plasma EV population hampers the development of new diagnostic tests. We hypothesize that the isolation of EVs with thyroid-specific surface molecules followed by miRNA analysis, may have improved diagnostic potency. Methods: The total population of EVs was isolated from the plasma of patients with FA (*n* = 30) and FTC (*n* = 30). Thyroid peroxidase (TPO)-positive EVs were isolated from the total populations using immune-beads. The miRNA from the TPO(+)EVs obtained from the plasma of FA and FTC patients was assayed by RT-PCR. The diagnostic potency of the selected miRNAs was estimated by the receiver operating characteristic (ROC) analysis. Results: TPO(+)EVs can be efficiently isolated by immunobeads. The analysis of Let-7 family members in TPO(+)EVs allows one to distinguish FA and FTC with high accuracy (area under curve defined by ROC = 0.77–0.84). Conclusion: The isolation of TPO(+)EVs, followed by RT-qPCR analysis of Let-7 family members, may present a helpful approach to manage follicular nodules in the thyroid gland.

## 1. Introduction

Incidences of malignant thyroid tumors are increasing worldwide. Follicular thyroid cancer (FTC) is the second most common type, accounting for about 5–10% of all thyroid malignancies. The courses of FTC differ from the course of most common papillary thyroid cancers. FTC affects younger people, has greater metastatic potency, and a higher mortality rate. Besides an aggressive clinical behavior, the management of FTC is challenged by a problematic diagnostic process [1]. The most important issue is distinguishing between truly malignant FTC and benign follicular adenomas (FA), which are encountered approximately 5 times more frequently. The thyroid node size or growth rate does not differ between FTC and FA [2]. Ultrasound elastography does not provide an ultimate diagnosis [3]. Due to the absence of cellular or nuclear atypia, cytological analysis of the material obtained by fine-needle aspiration can only reveal the follicular nature of the thyroid node but cannot rule out malignancy [4]. Thus, there is currently no method to confidently exclude a malignant diagnosis, which is considered as an indication for surgery. Moreover, a post-surgery histological diagnosis of FA does not guarantee a good prognosis [5]. This reveals the need for the development of a new, possibly objective, accurate method for the differential diagnosis of the follicular thyroid node.

Potential benefits of “liquid biopsies” in diagnosis, prognosis, and personalization of thyroid cancer therapy have been discussed recently [6]. Among different types of circulating markers, nano-scaled membrane extracellular vesicles (EVs) attract special interest [7]. The smallest vesicles of endocytic origin, like exosomes, are secreted by apparently all types of cells and reflect their biochemical properties and condition. Since circulating EVs present a natural, extremely stable, and cell-specific complexes of membrane lipids, proteins, and nucleic acids (mRNA, lncRNA, miRNA), they are considered as a promising marker for cancer diagnosis [8]. For example, a few years ago, we reported the diagnostic potency of exosomal miRNAs (miR-31, miR-21, miR-181a) for discrimination between benign FA, FTC, and papillary thyroid cancer (PTC) [9]. 

However, the development and clinical implementation of EV-based diagnostic tests are still hampered. One issue is an extreme heterogeneity of plasma EV populations with the majority of vesicles derived from the endothelium, platelets, blood, and stromal cells. A certain portion of circulating EVs may account for vesicles secreted by cells of specific tissues, e.g., the thyroid gland. Apparently, the fraction of tissue-derived EVs is defined by the number of specific cells, their secretory activity, and perhaps other factors. In the case of tumor development, the portion of malignant tissue-specific vesicles may increase. For instance, in an in vivo model of glioblastomas, tumor cell-derived exosomes accounted for 35–50% of the total population of circulating vesicles [10]. Analysis of the isolated fraction of tumor- or even tissue-specific vesicles would have greatly improved the diagnostic potency. Recently, this approach has been explored in the context of different types of cancer. For instance, EVs secreted by head and neck squamous carcinoma cells have been isolated via immune-binding of CD44v3 on their surface [11], while melanoma-derived exosomes have been captured via binding to CSPG4 [12].

By planning the presented study, we hypothesize that the isolation of thyroid cell-derived EVs, followed by the analysis of the vesicular miRNAs, may help to distinguish patients with FA and FTC. The hypothesis is based on the reported involvement of thyroid-derived EVs in thyroglobulin processing [13] and the detection of thyroid peroxidase (TPO) within vesicles [14,15]. Consequent steps of the study are schematically presented in Figure 1. After characterization of EVs isolated from plasma, the reduction of TPO-positive EVs (herein referred to as TPO(+)EVs) in the plasma of FA patients after a thyroidectomy was detected, confirming the thyroid origin of these vesicles. Next, TPO(+)EVs were isolated from the plasma of patients with FA and FTC and the vesicular miRNA profiled. Finally, the diagnostic potency of marker miRNAs (members of Let-7 family) in total and in the TPO(+) population of EVs was compared.

## 2. Materials and Methods

### 2.1. Patients

Patients with thyroid nodules were treated at the North-West Centre of Endocrinology and Endocrine Surgery and the N.N. Petrov National Medical Research Centre of Oncology, Saint-Petersburg. Evaluation of patients included ultrasound and fine-needle aspiration biopsy followed by a cytological exam. Patients who were classified as Bethesda category III-IV and were planned for thyroidectomy were enrolled in the study. Patients with multiple thyroid nodules and/or any endocrine or metabolic comorbidity were excluded from the study. Patients in whom the post-operative histological diagnosis did not confirm the follicular nature of the node or histological diagnosis was not unambiguous (FTC vs. FA), were excluded from the study as well. Relevant clinical characteristics of patients presented in Table 1.

The study was approved by the local ethics committee of N.N. Petrov NMRC of Oncology (10.11.2017, protocol AAAA-A18-118012390156-5) and conducted in accordance with the ethical guidelines outlined in the Declaration of Helsinki. All patients signed an informed consent form before being included in the study. Biological material and clinical data have been depersonalized. Plasma samples from patients with histologically confirmed benign follicular adenoma (*n* = 30) and thyroid follicular cancer (*n* = 30) were used. Additionally, the plasma of healthy female donors (*n* = 5) was obtained from the Department of Blood Transfusion of N.N. Petrov National Medical Research Centre of Oncology.

### 2.2. Plasma Sampling and EV Isolation

Blood samples were collected 3 times, namely, before the operation (a), after 5 days (b), and 4 weeks (c) after surgery. Peripheral blood was collected in vacutainers with EDTA (SARSTEDT AG & Co. KG, Nümbrecht, NO. 32.332). The plasma was separated by centrifugation within 10–15 min and stored at −80 °C. Further analysis proceeded with only plasma samples from patients with a histological diagnosis (FA or FTC) that was independently confirmed by two pathologists.

EVs were isolated from plasma by ultra-centrifugation (UC). Briefly, the samples (2 mL) were diluted by PBS 1:1, centrifuged at 12,000× *g* for 30 min at +4 °C. The supernatant was filtered using a 0.22-μm filter, then ultra-centrifuged at 110,000× g (Ultracentrifuge Optima XPN 80, rotor 70.1 Ti/k-factor 36) for 4 h at +4 °C. The pellet was washed once in 3.6 mL of PBS and then ultra-centrifuged again at 110,000 × g for 4 h at 4 °C. The pelleted vesicles were resuspended in 200 uL of PBS and kept at −80 °C. We have submitted all relevant data of our experiments to the EV-TRACK knowledgebase (EV-TRACK ID: EV200077) [16].

### 2.3. Analysis of Physical Characteristics of EV

Nanoparticle Tracking Analysis (NTA) was used to determine the size and the concentration of isolated vesicles. The measurements were carried out with a Nanosight NS300 analyzer (Malvern Panalytical Ltd., Malvern, UK) and the data obtained were processed by the Nanosight NTA 3.2 software package. Atomic force microscopy (AFM) was performed with a NT-MDT Solver Bio scanning probe microscope (Molecular Devices and Tools for Nanotechnology / NT-MDT, Moscow, Russia) in the tapping mode, using probe NSG01_DLC. The samples were applied to the mica surface (2SPI, West Chester, PA, USA), then immediately after that the upper layers were removed, incubated for 30 s, washed twice with distilled water, then dried using compressed air. Image processing was performed with the Gwyddion 2.56 (gwyddion.net) and Image Analysis (NT-MDT, Russia) programs.

### 2.4. Analysis of Biochemical Characteristics of EV (FACS, Dot-Blot)

In order to assay the presence of exosomal markers on the surface of vesicles, we used an Exo-FACS Kit (HansaBioMed, Tallinn, Estonia, NO. HBM-FACS-PEP). The EVs isolated by UC were incubated with 4 µm aldehyde-sulfate latex beads inPA accordance with the producer’s protocol. The tetraspanins CD63 and CD9 were detected with a monoclonal FITC-labelled antibody to CD63 (AbCam, Cambridge, MA, USA, ab8319) and a monoclonal PE-labelled antibody to CD9 (AbCam, Cambridge, MA, USA, ab82394), correspondently. Incubation was performed at a working concentration of antibodies 1 of mg/mL for 2 h at +4 °C in the dark. The complexes of “latex bead-EV-antibodies” were washed twice and diluted in PBS (200 uL). The samples were analyzed in a CytoFLEX (Beckman Coulter, Miami, FL, USA) flow cytometer equipped for multi-parametric and multi-color analysis, including a 488 nm argon laser for measurement of forward light scattering (FSC) and orthogonal scattering (SSC). The complexes assembled without EVs were used as a negative control. The samples were acquired and gated by FSC and SSC and then analyzed. FSC and SSC, as well as the fluorescence FITC and PE signals, were collected and stored as list mode files. The data were analyzed with the CytExpert (Beckman Coulter, Miami, FL, USA) software package.

In order to assay the presence of markers of interest in the whole vesicles, they were incubated with a 1x RIPA buffer solution (Thermo Scientific, Wilmington, DE, USA, NO.89900) for 10 min at +4 °C. The total amount of proteins was normalized by a Bradford assay, and 1 uL of the equilibrated lysate was spotted onto the nitrocellulose membrane. The membranes with spotted samples were then blocked with 5% BSA in TBS-T for 1 h at room temperature, then washed twice and incubated with the primary antibodies against TPO (AbCam, Cambridge, MA, USA, ab109383) or TSG-Rec (Santa Cruz, California, CA, USA, sc7964) diluted at 1:1000 in 0.1% BSA in TBS-T. Then, the membranes were washed with TBS-T and incubated with the HRP-labelled secondary antibodies (AbCam, USA-ab6789, ab6721), diluted at 1:10000 in 0.1% BSA in TBS-T for 30 min, then washed with TBS-T and evaluated with a Pierce ECL Western blotting substrate (AbCam, USA-ab133406) using the iBright FL1500 Imaging System (Invitrogen, Carlsbad, CA, USA). The received signal data were processed in the ImageJ program.

### 2.5. Isolation of TPO(+)EVs with Immune Beads

Superparamagnetic beads (SPMB) coupled with streptavidin (“Sileks Ltd.”, Moscow, Russia) and a biotinylated antibody to TPO (clon TPO28 “HyTest Ltd.”, Turku, Finland) were used. For this, 1 uL of the SPMB suspension (1 mg/mL) and 10 uL of the antibody (0.6 mg/mL) were mixed in 50 uL of PBS and incubated for 1 h at +4 °C with slow rotation. Then, the supernatants were removed using a magnetic separator, and the SPMB-antibody complexes were washed twice with PBS, then mixed with a suspension of EVs (50 uL) isolated from 0.5 mL of plasma. Incubation was carried out overnight at +4 °C with slow rotation. After incubation, the supernatants were placed into clean tubes and used for a second round of isolation. Complexes of SPMB-antibodies-TPO(+)EVs were washed by PBS and used for further experiments (flow cytometry, dot-blot, or RNA isolation).

In order to estimate the efficacy of TPO(+)EV attachment onto the SPMBs, complexes of SPMB-antibodies-TPO(+)EV were blocked with 0.2% I-Block Protein-Based Blocking Reagent, washed twice by PBS, and then incubated with the monoclonal FITC-labelled antibody to CD63 (AbCam, Cambridge, MA, USA, ab8319) for 2 h at +4 °C in the dark. Complexes were washed twice, diluted in PBS (200 uL), and then analyzed with a flow cytometer, as described above. Once the efficacy of the TPO(+)EV isolation was confirmed, the SPMB-antibodies-TPO(+)EV complexes were directly used for total protein or RNA isolation with a RIPA buffer or the Lyra reagent, respectively.

### 2.6. RNA Isolation and miRNA Profiling

RNA from the total population of EVs or TPO(+)EVs was isolated by the Lyra reagent (Biolabmix Ltd., Novosibirsk, Russia, NO. LRU-100-50) in accordance with the producer’s protocol, which was assumed to preserve small RNA molecules and microRNA. The concentrations and purities of the extracted RNAs were analyzed with an Implen NanoPhotometr N60 spectrophotometer (Implen Gmb, Munchen, Germany). The RNA concentration ranged from 20 to 130 ng/μL. The preparations were considered pure, with absorption rates of A260/A280 above 1.8. For the purpose of preliminary miRNA screening, 10 samples were chosen from patients with FA and 10 samples from patients with FTC. Aliquots of individual RNA samples were combined in equivalent quantities into 2 pools, representing FA and FTC. In the first step, the RNA was polyadenylated and reverse transcribed using a miRCURY LNA Universal RT microRNA polyadenylation and cDNA synthesis kit (Qiagen, Germantown, MD, USA, cat. number 339340). Then, quantitative PCR analysis was performed using miRCURY LNA miRNA Focus PCR Panels (Qiagen, USA, cat. number 339325) and an ExiLENT SYBR Green Master Mix (Qiagen, USA, cat. number 339345) with a CFX96 Touch™ Real-Time PCR Detection System (Bio-Rad, Hercules, CA, USA). Inter-plate amplification rate discrepancies were corrected with interpolated calibrators. Values of Ct higher than 38 were considered as background values and were excluded from the analysis.

### 2.7. Quantitative Real-Time RT-PCR (qPCR)

For the evaluation of selected miRNAs (Let-7 family) and miR-191 expression, miRNA-specific reverse transcription with stem-looped primers was used, followed by quantitative PCR. The IDs of the miRNAs, sequences of primers, and FAM-labeled probes were presented as Appendix A. Reverse transcription (RT) assays were performed using 1 uL RNA, a RT primers (50 pM) using a M-MuLV-RH RT kit (Biolabmix Ltd., Novosibirsk) with a 20 uL reaction volume. The reaction was carried out at 25 °C for 45 min, followed by incubation at 85 °C for 5 min to inactivate the reverse transcriptase. Quantitative PCR analysis was performed using a 3 uL RT reaction mix, PCR primers (0.3 uM for both), FAM-labeled probe (0.2 uM) with a BioMaster HS-qPCR (2×) kit (Biolabmix Ltd., Novosibirsk). A CFX96 Touch™ Real-Time PCR Detection System (Bio-Rad, USA) was used for all reactions. The conditions for qPCR were as follows: Temperature of 95 °C for 10  min and 45 cycles of 95 °C for 5  s, followed by 65 °C for 15  s. RT-qPCR reactions for each miRNA molecule were repeated independently twice and then averaged. The results of the RT-qPCR were normalized to the total mean of cycle threshold (Ct) and to the reference Ct of miR-191 independently. The obtained data were analyzed with a CFX Manager, SigmaPlot 11.0, and GraphPad Prism 8.0.2.

### 2.8. Statistics

Statistical evaluation of the differences between comparable groups of samples was done with Mann-Whitney or Kruskal Wallis tests assuming a non-parametric distribution of measured parameters.

## 3. Results

### 3.1. Isolation and Characteristic of EV Population

The total population of plasma EVs was isolated by the standard procedure of differential UC. The concentrations and sizes of isolated particles were determined by NTA in each sample included in the analysis. The concentration varied from 1.5 to 3.6 × 10^8^ particles/mL. A representative example ((3.43 ± 0.3) × 10^8^ particles/mL) is shown in Figure 2. Before NTA, the nano-vesicles isolated from 2 mL of plasma were resuspended in 200 uL of PBS (concentrated 10 times), then 1 uL of the vesicle suspension was diluted up to 1000 mL of PBS (dissolved 1000 times). Considering these manipulations, the initial concentrations of nano-vesicles in plasma were in a range of 1.5 to 3.6 × 10^10^ particles/mL. This parameter did not reveal any correlation between the gender or age of patients, size of the thyroid nodule, or post-operative histological diagnosis. The morphologies of the isolated vesicles were analyzed for several samples by AFM (Figure 3A). The size distribution was detected in a range of 40 to 120 nm, with a major fraction in a range of 60 to 80 nm, as shown in the histogram in Figure 3B. The expressions of exosomal markers (tetraspanins CD9 and CD63) on the surface of isolated vesicles were assayed by flow cytometry (Figure 4). Considering the size, morphology, and presence of “exosomal” surface markers, the isolated nano-vesicles were assumed to be presented, at least partially, by exosomes.

### 3.2. Analysis of TPO(+)EVs and TSG-Rec(+)EVs in Circulation after A Thyroidectomy

Thyroid peroxidase (TPO) is an enzyme involved in the synthesis of thyroid hormones, and its expression is enriched in follicular cells of the thyroid [17]. The presence of TPO in thyroid cell-secreted exosomes has been demonstrated recently [15], as well as the association of circulating TPO(+) vesicles with thyroid gland disorders [14]. To confirm this association, we assayed the presence of TPO in EVs isolated by UC from plasma samples (*n* = 10) obtained before, 5 days after, and 4 weeks after a thyroidectomy. In parallel, the same samples were tested for the presence of thyrotropin receptors (or TSH-Rec), the expression of which is limited by the cells of the thyroid gland. Representative examples of dot-blots are shown in Figure 5A and the results averaged for 10 patients are presented in Figure 5B. Thus, surgery resulted in a temporal, statistically non-significant increase of thyroid-derived EVs in circulation, while the concentration of these vesicles was significantly reduced 4 weeks after the operation. These data confirm the thyroid origin of the TPO(+)EVs and TSG-Rec(+)EVs.

### 3.3. Preparative Isolation of TPO(+)EVs

It can be assumed that TPO(+) thyroid-derived EVs account for a minor fraction of circulating vesicles and that their input in the miRNA profile of plasma EVs is negligible. This would compromise the potency of any diagnostic test based on the analysis of the total population of plasma EVs. We attempted to isolate thyroid-derived EVs using superparamagnetic beads (SPMB) decorated by anti-TPO antibodies. A workflow of the experiment is presented schematically in Figure 6. After coupling the SPMBs with antibodies via streptavidin-biotin (Figure 6A), the SPMB-antibodies complexes were incubated with the total population of plasma vesicles, resulting in capturing TPO(+)EVs (Figure 6B,C). In order to confirm the efficacy of the TPO(+)EV isolation, flow cytometry and dot-blot analysis was used.

The development of follicular nodules in the thyroid gland is associated with an enlargement of the size of the gland and, we suppose, an increase of the number of TPO(+)EVs in circulating plasma. To test the efficacy of proposed technology of TPO(+)EVs isolation, we used the plasma samples of FA patients (*n* = 5) with the largest size thyroid gland as it was estimated by ultrasound (>130 mL) and healthy donors (*n* = 5). It was explored whether the samples of the first group had an enriched amount of TPO(+)EVs, and efficacy of these vesicles isolation was higher compared to the second group. Beads coupled with anti-TPO antibodies were incubated in parallel with samples of EVs isolated by UC from the plasma of FA patients and healthy donors. The captured EVs were labeled by the antiCD63-FITC antibody and assayed by flow cytometry. Representative results are presented in Figure 7A–C. A negative control (beads not incubated with EV) was used to set a threshold of fluorescent signal intensity assuming 0.83% of positively counted events as a background. In sample of healthy donor, 2.19% of SPMBs were positive and, therefore, carried TPO(+)EVs, whereas for the sample of the FA patient, the amount of positive SPMBs reached 13.67%. To test if the applied method indeed resulted in TPO(+)EV capture by SPMBs and the depletion of TPO(+)EV from suspension, the procedure was repeated using the supernatant after the first round of isolation. The results are shown in Figure 7D,E. The amount of TPO(+)EVs decreased from 2.19% to 1.27% in the control sample and from 13.67% to 1.23% in the sample from the FA patients. Next, assay was repeated with samples from four other healthy donors and four FA patients with a similar size of thyroid gland and clinical status (Table 1). The averaged results demonstrate the reproducible character of our observations (Figure 7F).

To further confirm the efficacy of TPO(+)EVs isolation, we assayed the relative amount of TPO in samples during the procedure of isolation by dot-blotting. The 1 uL initial suspension of EV (1), 1 uL of the supernatant (2), and all amounts of SPMP-antibodies-TPO(+)EV complexes (3) obtained after isolation were mixed with a RIPA buffer in a final volume of 5 uL. Ten samples were spotted in parallel onto two membranes and then evaluated for TPO and CD9 (Figure 8A). Both TPO and CD9 were presented in samples of the initial EV suspension (upper line: 1, EV suspension) and the supernatant after the isolation of TPO(+)EVs (middle line: 2, supernatant). TPO was clearly detected in the SPMP-TPO(+)EV complexes (bottom line: 3, SPMP-TPO(+)EV), while the amount of CD9 was hardly detectable. Assuming CD9 as one classic “exosomal marker”, we normalized the signal intensity from TPO versus CD9 and averaged the data for 10 samples (Figure 8B). The ratio of intensity for TPO/CD9 in samples of SPMP-TPO(+)EV was considerably higher than in the initial suspension of EVs or in the supernatant after the isolation of TPO(+)EV. Thus, two different approaches were used to demonstrate the efficacy of the proposed technique for TPO(+)EVs isolation.

### 3.4. MiRNA Profiling of TPO(+) EV from Plasma of FA and FTC Patients

In the next experiments, we wanted to test the hypothesis that the miRNA content of follicular cell-derived EVs may help to distinguish between patients with benign and malignant follicular nodes in the thyroid gland. Samples of plasma taken before surgery were used. RNA was isolated from the TPO(+)EVs of FA patients (*n* = 10) and FTC patients (*n* = 10). To perform preliminary profiling of 85 cancer-associated miRNAs, we combined RNA samples in an equivalent quantity into two pooled samples, representing FTC and FA. The RNA in both samples was poly-adenylated, reverse transcribed with polyT-primers, and diluted cDNA was used for quantitative PCR with a miRNA-specific primer in a 96-well plate. Inter-plate calibrators resulted in an almost equal Ct value (UniSp3 IPC of 26.3 and 26.6), indicating an equivalent RNA quality between the two pooled samples. Values of Ct higher than 38 were considered as background values and were excluded from the analysis. After the correction of inter-plate discrepancy, the results were normalized to the global Ct mean and log2 transformed (Figure 9A). The microRNA contents of TPO(+)EVs from FA and FTC patients differed significantly. Moreover, from 85 molecules included in assay, 27 miRNAs were detected in both samples for FA and FTC. Interestingly, six miRNA of the Let-7 family (Let-7a, Let-7b, Let-7d, Let-7f, Let-7g, Let-7e) were detected in both samples and were over-represented in TPO(+)EVs from FTC plasma samples when compared to FA. The involvement of Let-7 in thyroid carcinogenesis has been described in a number of studies [18], however, all reports published thus far have focused on papillary thyroid cancer. For example, the elevated level of Let-7a, Let-7c, Let-7d, and Let-7f in the plasma of papillary thyroid cancer patients has been reported recently [19]. Moreover, Let-7 is being discussed as a promising serum for miRNA markers for the diagnosis and prognosis of thyroid cancer [20]. Considering these data, we decided to further focus on the analysis of Let-7 miRNAs in the plasma EVs obtained before surgery. Six members of the Let-7 family were assayed in the total population of EVs and PTO(+)EVs from the plasma of FA (*n* = 30) and FTC (*n* = 30) patients using miRNA-specific reverse transcription with stem-looped primers and qPCR (Figure 9B). In the case of TPO(+)EVs, four miRNAs, namely, Let-7b, Let-7d, Let-7f, and Let-7g, were significantly over-represented in FTC samples when compared to the FA samples. These results were observed by direct comparison of the Ct values and observation of the values after normalization to the averaged Ct values of all six tested Let-7 miRNAs or to miR-191 as established plasma miRNA reference (not shown). When the miRNA was analyzed in the total population of EVs, over-representation of the tested miRNA in FTC samples was not detected. Even the amount of miR-7d was lower in EVs from FTC when compared to the FA samples. Thus, the analysis of miRNAs isolated from TPO(+)EVs reveals the over-representation of Let-7b, Let-7d, Let-7f, and Let-7g, which was not detectable by analysis of the total EV population.

### 3.5. Evaluation of Diagnostic Potency of Let-7 miRNAs from EV

In order to estimate the diagnostic potency of Let-7 in different populations of plasma EVs, ROC analysis was applied for the RT-PCR data obtained for Let-7a, Let-7b, Let-7d, Let-7f, Let-7g, Let-7e with 60 samples (*n* = 30 for FA and *n* = 30 for FTC). The results are presented in Figure 10. In four cases, the analysis of marker miRNA from the TPO(+)EVs had a higher diagnostic potency when compared to the analysis of miRNAs from the total population of circulating EVs. It is interesting to note that the result of the group comparison (Figure 9B) was correlated with the ROC analysis (Figure 10). A slight difference of Let-7d representation in the FTC and FA groups was associated with a low value of the UAC (0.647 and 0.779, respectively), while a significant difference of Let-7f representation in the FTC and FA groups was associated with a high value of UAC, especially in PTO(+)EVs (0,683 and 0.814, respectively).

On the basis of the obtained results, we conclude that TPO(+)EVs and their components may present promising markers for the liquid biopsy-based differentiation of FA and FTC. Speaking more generally, the power of EV-based diagnostic approaches will definitely be of benefit when these approaches are implemented for the analysis of tissue- or cell-specific EVs.

## 4. Discussion

Thyroid cancer’s mortality has increased over the last years that indicates the need to develop new diagnostic approaches. The advantages of new immuno-histochemical (Hector Battifora mesothelial cell 1 (HBME-1), high molecular weight cytokeratin 19 (CK19), galectin-3, c-met) and genetic (TERT, BRAF, PAX8/PPARγ, RAS, and RET/PTC) markers to be tested in the material of cytological or histological biopsy are discussed explicitly [21,22,23]. The development of the noninvasive diagnostic tools for analysis of the circulating plasma components, also known as “liquid biopsy”, brings a new insight for thyroid cancer diagnosis. Thus, traditional markers of thyroid cancer, like BRAF mutation or miRNA expression profile, can be evaluated in plasma [24]. The circulating DNA methylation pattern of some iodine transporters and DNA methyltransferase were tested as a promising diagnostic biomarker of thyroid cancer as well [25]. The list of candidates for “liquid biopsy” markers of thyroid cancer includes circulating tumor cell (CTC), exosomes, and extrachromosomal circular DNA (ecDNA) [6], that indicates active research in this area. Exosomes, important mediators of intercellular communications involved in the development of various thyroid disorders, including auto-immune pathologies [14] and cancer [26], are considered to present a new marker for liquid biopsy application.

The first investigations proposing the diagnostic value of circulating extracellular nano-vesicles or exosomes were published many years ago [27]. Since then, hundreds or even thousands of studies relevant to vesicular markers of different types of cancer have come out. Diagnostic potency of plasma exosomes in thyroid cancer has been explored in several studies [9,28]. However, EV-based methods for cancer diagnosis, prognosis, or management, are still far from clinical implementation. This is caused by obvious technological issues related to EV isolation and analysis. Confident results are still hardly achievable, even with ISEV’s regular efforts to standardize research in this area [29,30]. Besides methodological aspects, important biological issues need to be addressed before circulating EVs can be used as feasible cancer diagnostic markers. First of all, the population of circulating nano-vesicles is too complex and heterogenous in terms of the cellular origin to reflect the condition of certain tissues, for example, when focusing on malignant transformation. The development of an approach to isolate fractions of EVs released by cells of certain tissues presents an extremely important issue.

In the presented study, we have attempted to isolate thyroid-derived EVs using a single surface marker, thyroid peroxidase, which is expressed by thyroid follicular cells. Our result reveals the feasibility of isolating TPO(+)EVs from only 2 mL of plasma in an amount sufficient for further analysis. The proposed strategy is very promising because it can be applied for the diagnosis of other types of thyroid cancer or even other thyroid pathologies. However, several methodological issues should still be considered. First, the expression of TPO is not absolutely restricted by the thyroid gland [31], and a fraction of the EVs isolated by the proposed technology is enriched by (rather than consists of) thyroid-derived vesicles. The efficacy of the thyroid-derived EV isolation can be optimized by using the ligands to other thyroid-specific molecules. For instance, The protein atlas provides 13 thyroid-enriched genes, including TPO and TSH-Rec [17]. Moreover, it can be assumed that thyroid-derived vesicles present a miserable fraction of the total population of plasma EV. Thus, the optimal ratio between the specificity and the efficacy of the thyroid-derived EV isolation will result in the necessary purity and quantity of the isolated vesicles. This issue needs to be addressed experimentally.

The list of analyzed components of thyroid-derived EVs present one more disputable aspect. In the presented study, we have analyzed vesicular miRNAs only, and our profiling assay was limited by only 85 cancer-associated miRNAs. This definitely restricts the diagnostic potency of the proposed method. In addition to the Let-7 miRNA family, FTC-relevant miRNAs, other types of vesicular RNA, or other vesicular components can serve as a marker of follicular cell malignancy. Moreover, an increase in the plasma level of Let-7 miRNAs was also associated with papillary thyroid cancer [19]. These data and the design of our study do not allow us to argue about the diagnostic specificity of Let-7, even isolated from TPO(+)EV. Thus, the diagnostic potency of the proposed method can be augmented by the analysis of larger lists of molecules associated with particular thyroid pathologies.

Even considering the limitations mentioned above, the proposed approach allowed us to discriminate FTC from FA with quite a high accuracy (AUC = 0.81). The results obtained are still not reliable enough to be recommended for practical implementations, however, they confirm the validity of the proposed analytical approach and provide consolidation for further research. Diagnostic potency of the described method can be obviously improved by the use of multiple thyroid-specific ligands for EVs isolation and by the analysis of various EV components associated with specific thyroid diseases.

## Figures and Tables

**Figure 1 cells-09-01917-f001:**
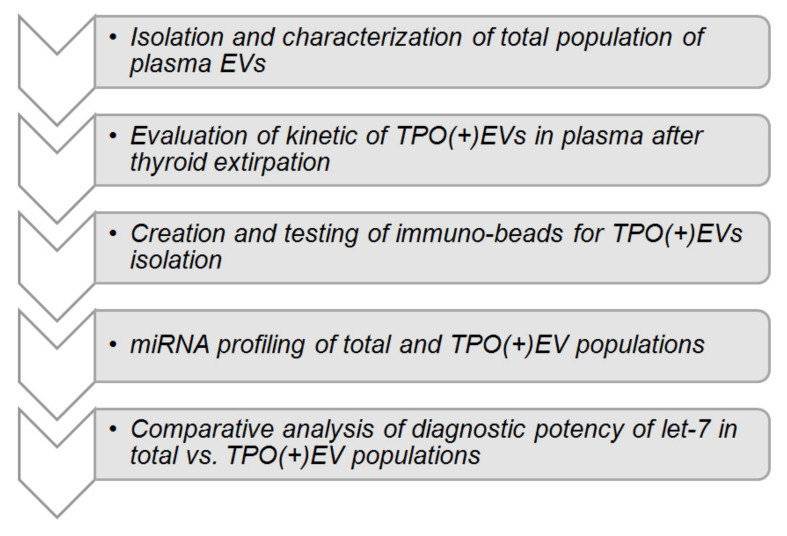
Study design.

**Figure 2 cells-09-01917-f002:**
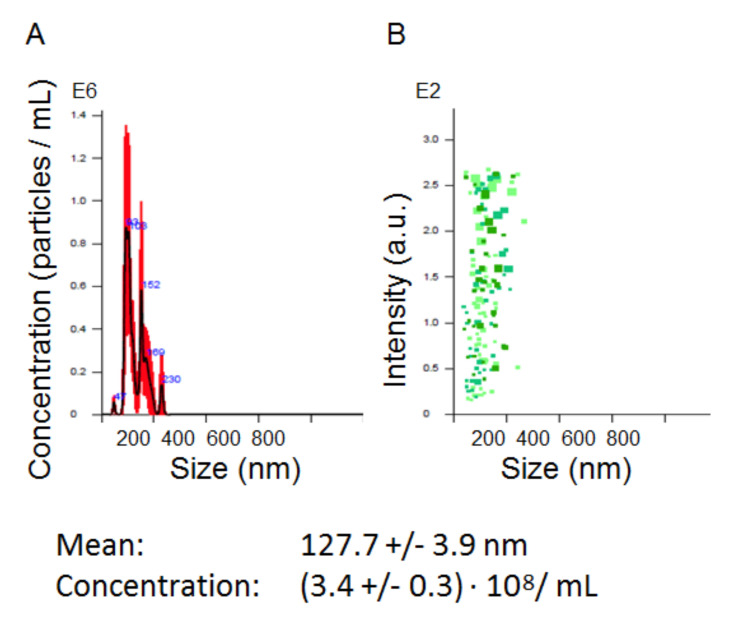
Representative example of the nanoparticle tracking analysis (NTA) of extracellular vesicles (EVs) isolated by ultra-centrifugation (UC). (**A**) Finite track length adjustment (FTLA)/size graph. (**B**) Intensity/size graph.

**Figure 3 cells-09-01917-f003:**
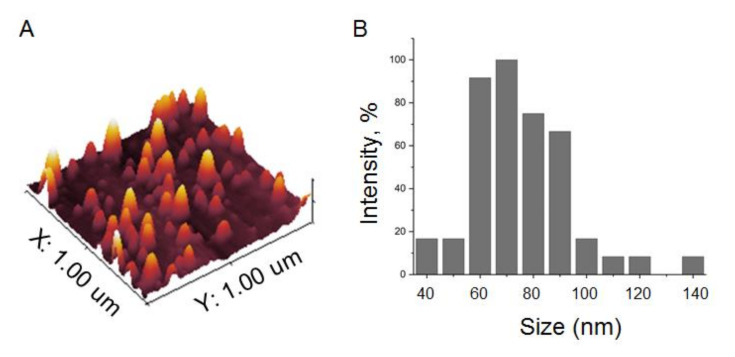
Atomic force microscopy of EVs isolated by UC. (**A**) Topography of a scanned sample surface. (**B**) Histogram reflecting the size distribution of vesicles attached to the scanned surface.

**Figure 4 cells-09-01917-f004:**
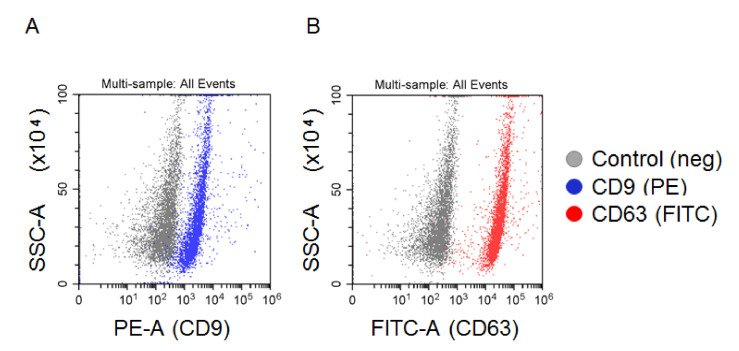
FACS analysis of EVs attached to latex beads. EV vesicles attached to latex beads were stained with PE-labelled antibodies to CD9 (**A**) or FITC-labelled antibodies to CD63 (**B**). Beads without EVs stained with antibodies were used as a negative control in each experiment.

**Figure 5 cells-09-01917-f005:**
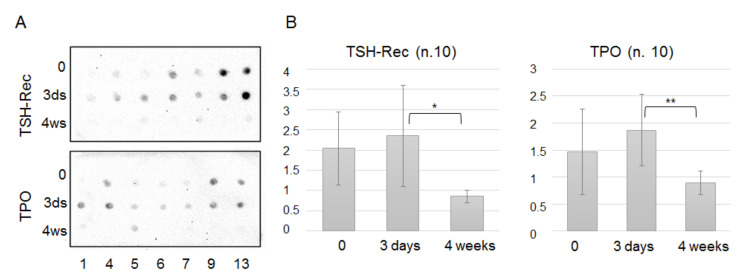
Dot-blot analysis of thyroid peroxidase-positive (TPO(+)) and thyrotropin receptor-positive (TSH-Rec(+)) EVs in the circulation of patients. Plasma samples were taken 3 times, namely, before the thyroidectomy, 5 days after the thyroidectomy, and 4 weeks after the thyroidectomy. EVs were isolated by UC. Samples were normalized vs. the total protein amount, spotted onto a membrane, and then evaluated with primary antibodies against TPO or TSH–Rec and an HRP-labeled secondary (**A**). The integrated values of the spot sizes and intensities were generated by the ImageJ program and averaged for 10 samples (**B**). The statistically significant differences between paired groups (“0” vs. “5 days” and “5 days” vs. “4 weeks”) were evaluated by the Mann–Whitney test and indicated as * (*p* < 0.5) and ** (*p* < 0.005)

**Figure 6 cells-09-01917-f006:**
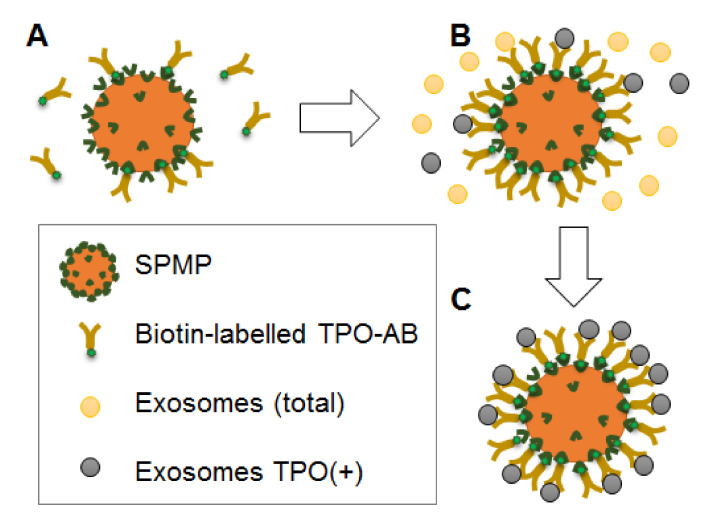
Schematic work-flow of TPO(+)EV isolation. (**A**) Coupling of anti-TPO antibodies with super-paramagnetic particles via biotin-streptavidin binding. (**B**) Incubation of SPMP-anti-TPO antibodies complexes with the total population of plasma EVs and the capturing of TPO(+)EVs. (**C**) Isolation of specific populations of TPO(+)EVs.

**Figure 7 cells-09-01917-f007:**
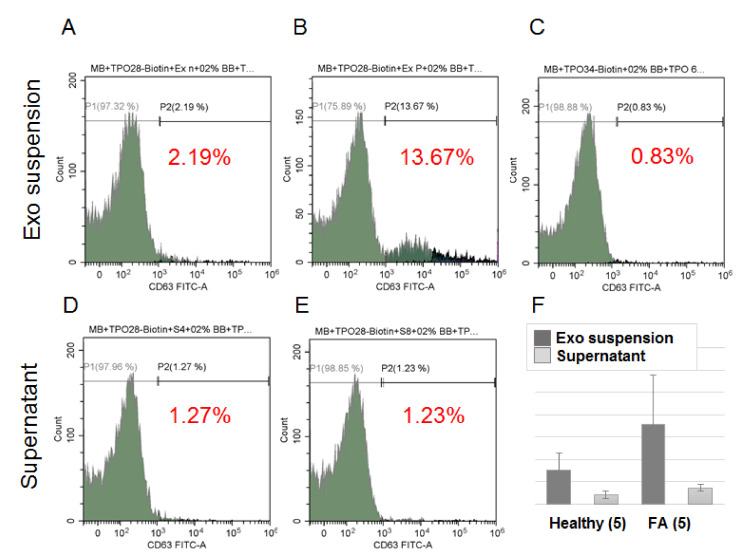
FACS analysis of efficacy of TPO(+)EV isolation. Complexes of SPMP-TPO(+)EV were stained with FITC-labelled antibodies against CD63. The upper panel of the histogram includes the result of the analysis of the SPMP-TPO(+)EV complexes isolated from the plasma of healthy donors (**A**), FA patients (**B**), and the control samples, presenting SPMP-anti-TPO AB complexes without EVs (**C**). The percentage of positive particles was calculated using the same threshold value. The lower panels of the histogram (**D**,**E**) include results of the analysis of the SPMP-TPO(+)EV complexes obtained after incubation with the supernatant after the first round of isolation (**A**,**B**, respectively). (**F**) Averaged results obtained from samples of healthy donors (*n* = 5) and patients with large follicular adenomas with a volume of the thyroid gland obtained by US > 130 mL (*n* = 5). The observed difference between paired groups of samples (EVs suspension vs. supernatant) was statistically significant (*t* < 0.05) as evaluated by the Mann–Whitney test.

**Figure 8 cells-09-01917-f008:**
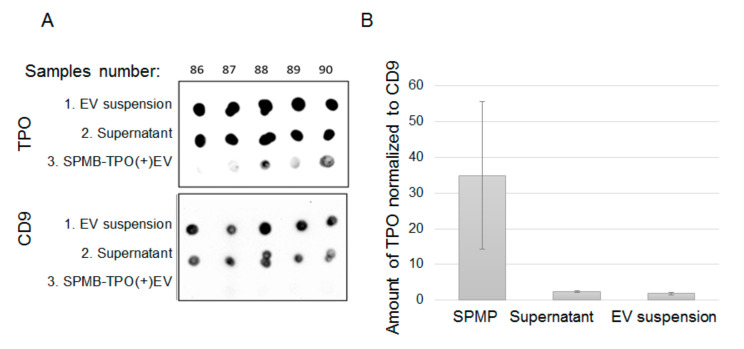
Dot-blot analysis of the efficacy of TPO(+)EV isolation. (**A**) Blots with similar samples (upper line: Total EV population; middle line: Supernatant TPO(+)EV-depleted; bottom line: SPMP-TPO(+)EV complexes) stained with antibodies against TPO and CD9. Blots included samples from 5 patients. (**B**) The result of 10 samples for dot-blotting were quantified using ImageJ. The ratios of TPO/CD9 signal intensities were calculated and averaged in three groups, presenting SPMP-TPO(+)EV complexes, supernatant TPO(+)EV-depleted, and the total EV population.

**Figure 9 cells-09-01917-f009:**
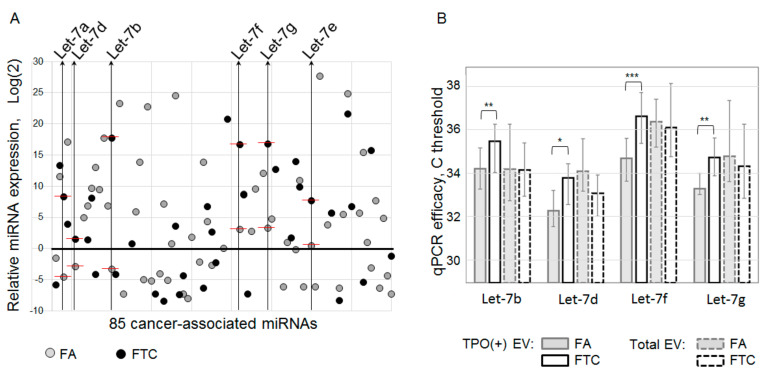
Cancer-associated miRNA expression profiling. (**A**) Analysis of two pooled samples combined from the RNA isolated from TPO(+)EVs of FA (*n* = 10) and FTC (*n* = 10) patients. Samples were assayed using a Cancer Focus RT-qPCR Panel, and the results were normalized to the global Ct mean and then log2 transformed. The results selected for further analysis of miRNAs the Let-7 family are marked and indicated. (**B**) Four miRNA members of the Let-7 family were assayed in the total EV population and TPO(+)EVs from individual samples. The results (Ct) were averaged for groups of FA (*n* = 30) and FTC (*n* = 30). The statistical significance of observed differences was evaluated by the Mann–Whitney test and indicated as * (*p* < 0.05), ** (*p* < 0.005), *** (*p* < 0.0005).

**Figure 10 cells-09-01917-f010:**
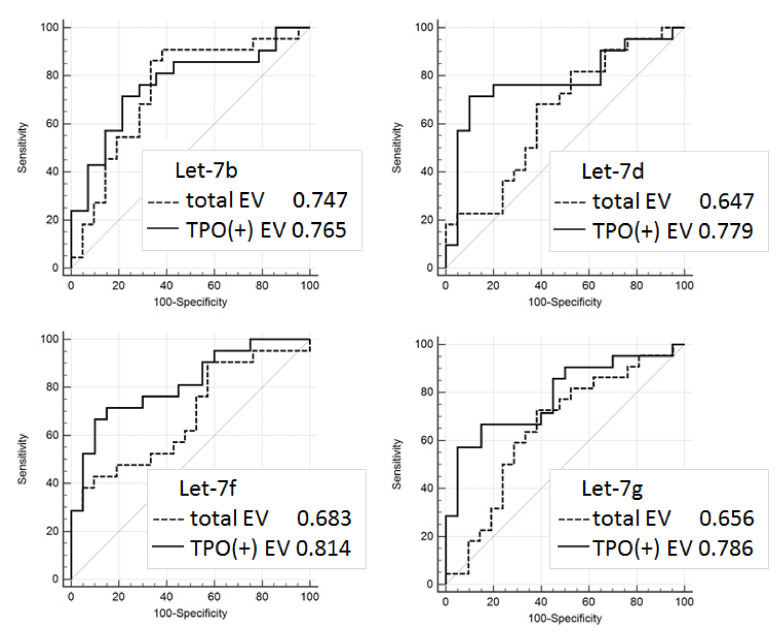
Diagnostic signature of Let-7 miRNA isolated from the total population of EV and TPO(+)EV. Receiver operating characteristic (ROC) curves for Let-7b, Let-7d, Let-7f, and Let-7g were created using the results from 60 samples: FA (*n* = 30) and FTC (*n* = 30) using the Graph Pad Prism software package.

**Table 1 cells-09-01917-t001:** Clinical characteristics of patients.

Abbreviation	Size of Thyroid Averaged, mL (Ultrasound Before Operation)	Age Averaged (years)	Gender (F/M)	Type of Thyroid Node (Histology After Operation)	Number
Healthy donors	16.8 (±2.2)	52.8	5/0	-	5
FA *	61.6 (±44.1)	51.4	21/9	Follicular adenoma	30
FTC	57 (±32.5)	54.5	26/4	Follicular thyroid cancer	30

* Group of patients with follicular adenoma (FA) includes 5 individuals with a large size of thyroid gland estimated before surgery by ultrasound (>130 mL). Samples of their plasma and plasma of healthy donors were used for the analysis of the efficacy of thyroid peroxidase(+)extracellular vesicles (TPO(+)EV) isolation.

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
