# Peer review of "MiRNA let-7 from TPO(+) Extracellular Vesicles is a Potential Marker for a Differential Diagnosis of Follicular Thyroid Nodules"

_cells, 2020, doi:10.3390/cells9081917_

Round 1

Reviewer 1 Report

        COMMENTS

The mnuscript titled "MiRNA let-7 from TPO(+) extracellular vesicles is a potential marker for a differential diagnosis of follicular thyroid nodules" of Zabegina L et al., has investigated benign and malignant thyroid lesions such as follicular adenomas (FA) and follicular thyroid cancer (FTC).

By liquid biopsy-based analysis of circulating extracellular vesicles (EVs), thyroid peroxidase (TPO)-positive EVs were isolated from the plasma of FA and FTC patients. Further, miRNAs from the TPO(+)EVs were assayed by RT-PCR.

The analysis of Let-7 family members in TPO(+)EVs showed significantly differences between two study groups.

Abstract:

The abstract is adequately describing this study. However,

At le line 26: before AUC acronym to add the text from which it derives.

Introduction:

The introduction is adequately describing the aims of study. However,

At the line 54: before FPC abbreviation to add a text such as follicular papillary thyroid carcinoma or better, papillary thyroid carcinoma (PTC).

At the line 77: see comment for line 26. AUC acronym is only valid by correction at line 26.

Materials and Methods:

Even if an inadequate number of cases has been tested, the parameters adopted to pick up patients are in line with thyroid histological criteria.

Clinical parameters of patients are absent or inadequately displayed.

The methodologies are appropriate. Further,

At the line 103: should be removed (NTA, AFM) because of these are acronyms of any previous text.

At the line 155: after "Lyra reagent" should be added "(Biolabmix 157 Ltd, Novosibirsk, № LRU-100-50)". Therefore, this specification should be removed from line 157.

At the line 170: before "ct" should be added "cycle threshold".

Results

The results are adequately described. However,

At the line 189: Ultra-centrifugation should be replaced by (UC) since this abbreviation has already been used at line 96.

At the line 198: it there is no need to write "(FA or FTC)" since this study has exclusively examined patients harbouring FA and FTC histological lesions.

At the line 228: "patients with a large follicular node (Bethesda III, volume of thyroid gland obtained by US >130 mL" have to be described in Materials and Methods section. The number of patients appears only in Legenda of Figure 6 "US >130 mL (n = 5)", at line 338. Farther, US acronym refers to any text.

At the line 230: I would be interested to know on what basis 5 of 30 FA patients will be selected.

At the lines 239-240: "From four other healthy donors and four FA patients with a similar clinical status." ?? To have a clear understanding of these data, the clinical features of patients should be displayed in Materials and Methods section or conversely, an adequate table could be useful to pick up clinical and histological reports.

At the line 375: should be removed "in our opinion", since it is pleonastic.

Discussion:

The comments of results are too short. This section could be enriched by expanding the comments on role of biomarkers on thyroid cancer delineated in "What is New on Thyroid Cancer Biomarkers" article.

Bibliography/References:

The bibliography is too short.

Most references are less than ten years old and they underline the growing importance of nanoparticle in thyroid gland. Specially, environmental nanoparticles have been associated with thyroid cancer.

Decision:

This study may be accepted for publication after revisions.

Author Response

Thank you very much for careful consideration of our work and important remarks!

All mistakes and spellings in Abstract, Introduction, Methods/Material and Results section are corrected.

New paragraph is added to explain reason for experiment with 5 samples of donor’s plasma and 5 samples of plasma from FA patients with large thyroid gland. Samples of patients were selected (5 from 30) on the base of thyroid gland size evaluated by ultrasound before surgery. We suggested that patients with largest volume of thyroid gland (due to follicular nodules development) could have an increased amount of follicular cell –derived (TPO(+)) EVs. These considerations were included into text.

Clinical parameters of patients are summarized in the Table 1 and included into manuscript.

Discussion part is expanded, several new articles focused on development of new tissue markers as well as potential liquid biopsy markers of thyroid cancer are cited.

We are appreciating your input to improve quality of our manuscript!  

Reviewer 2 Report

The manuscript by Zabegina Lidia et al. describes that let-7 from TPO(+) extracellular vesicles is a potential marker for a diagnosis of FTC.

I. Strong points of the article 

  1. The submitted study raises important issues and trendy topic. 

  1. The thyroid cancers present a limited number of genetic aberrations for ctDNA based liquid biopsies. 

  1. The idea is very interesting and could be inspiring for other authors.

  1. Despite many problems, the study design and methodology seems to be modern and original. However such ambitious studies requires very cautious approach.

II. In my opinion, at a current stage article requires many corrections or/and explanations.

Firstly, I have some questions which were not clearly explained

  1. Does the submitted study as the first reports MPO positive EV I thyroid cancer (position 15 in literature is in autoimmune disease), and overexpression of let-7 in exosomes in thyroid malignancies?

  1. Can this method (MPO+ EV) theoretically might be applied for any thyroid cancer including more common PTC?

  1. Are there any doubts about methodology or any risk of technical bias? If yes, this should be discussed. 

III. Major problems:

1. TPO is NOT exclusively expressed only in thyroid cancer, but also in the breast, and this issue needs to be carefully discussed (read below).  

Godlewska M, et al. Localization of key amino acid residues in the dominant conformational epitopes on thyroid peroxidase recognized by mouse monoclonal antibodies. Autoimmunity. 2012;45(6):476–84.  

2. Some question about Quantitative Real-Time RT-PCR (qPCR) a) How the probes were designed or they were applied previously?

a) If they designed how those were tested? They should be if this not commercial products.

b) Reference Ct of miR-191 – why this one? especially in the context that previously it has been reported that it might be down-regulated in thyroid? This needs to be explained, see the article below.

M Colamaio et al. 2011 miR-191 down-regulation plays a role in thyroid follicular tumors through CDK6 targeting.

3. The statistical analysis details are missing. Sample distribution P/NP, tests applied, furthermore correlations: which test was applied Spearman or Pearson’s?

4. In my opinion, the structure of the article is very unusual:

a) The introduction has some parts with results description (line 76-78)

b) Methods were mix with results in example in lines 260-266

c) Why the figures and tables are an additional subsection of results section “3.6. Figures, Tables and Schemes”?

d) The results are discussed in the results section (205-208, 270-275 lines) what shorts the discussion to just part of the page and includes just three citations 21-23?!?

And so on….

IV. Minor problems

  1. The table with Oligonucleotides used for the RT-qPCR analysis of miRNA might be easily added to supplementary materials? 

  1. As a potential reader, I really like the study design flow chart and some final remarks or conclusions, however I leaving this, up to the author's decision. 

  1. In figure 9 Receiver operating characteristic (ROC) curves present AUC I guess? (would be beneficial to see exact sensitivity and specifity)

  1. In the discussion, there are some research articles in my opinion that are missing suggesting the potential application of liquid biopsy in thyroid cancer diagnostics since TPO EV might also be used in other types of thyroid cancer?

E Perdas et al 2018 Potential of Liquid Biopsy in Papillary Thyroid Carcinoma in Context of miRNA, BRAF and p53 Mutation

or published very lastly very comprehensive review

Yi Wang et al. 2020, Exosomes as Mediators of Cell-to-Cell Communication in Thyroid Disease

  1. The table with patients' characteristics would be a good idea. Moreover, inclusion/ exclusion criteria are needed.

  1. First of all, I don't feel qualified to judge about the English language and style. However, in my opinion some mistakes of the manuscript I have noticed should be corrected to improve the quality of the text:

Line 23: by receiver operating characteristic shouldn’t be by the receiver…?

Line 40: Due to absence of cellular, shouldn’t be due to the absence?

Line 51: cell-specific complex of -  shouldn’t be complexes

Line 51: lnRNA: Long non-coding RNAs are more commonly written as lncRNA

In line 54: the reader can only guess what is FPC… lack of  abbreviation expansion

line 55: diagnostic tests is still: shouldn’t be tests are

line 59: by the quantity of specific cells: wouldn’t be easier number?

line 62: of total: of the total

line 76: … The receiver operating characteristic

line 82: … patients instead patents

line 107: shouldn’t be an NT-MDT?

line 157: TPO(+)EVs was isolated – were

Line 176: a RT primes – primers

Line 213: presented on Figure – in

Line 230: labelled – in am. Eng. labeled?

Line 254: the proposed?

Line 368: Confident and reproducible results are – “sentence is wordy”

Line 370: Beside methodological aspects – besides

 Line 372:  heterogeneous or heterogenous?

Line 373: when focusing… on missing

Author Response

Dear Reviewer!

Thank you very much for careful reading of our manuscript and important remarks!

We made requested changes in the text, added Figure 1 with scheme of study and Table 1 with clinical data of patients, and expanded discussion. All questions and problems are addressed below.

I Questions:

1. Does the submitted study as the first reports MPO positive EV I thyroid cancer (position 15 in literature is in autoimmune disease), and overexpression of let-7 in exosomes in thyroid malignancies?

We did not find any indications on implication of TPO(+)EV in thyroid cancer. Our report can be a first one. However, we assumed that follicular cells of thyroid as well as cells of differentiated thyroid cancer may produce vesicles with tissue-specific surface markers like TPO. We did not considered this phenomenon as cancer-specific event, therefore we did not focus attention on role of TPO(+)EV in thyroid cancer pathogenesis.

Function of let-7 miRNAs family is well studied in context papillary thyroid cancer. There are couple of articles that demonstrated decreased expression, tumor-suppressive role of these molecules and their link to activation of HMGA2 (Li et al., 2017; Perdas, Stawski, Nowak, & Zubrzycka, 2016; Ricarte-Filho et al., 2009). In contrast to down-regulation into follicular cells, four miRNAs (let-7a, let-7c, let-7d, and let-7f) were shown to be significantly up-regulated in plasma of PTC patients comparing to healthy controls (Perdas, Stawski, Kaczka, & Zubrzycka, 2020). That indicates possible reciprocal character of let-7 miRNAs concentration in PTC tissue and circulating plasma. Similar (tumor suppressive) role of Let-7a in follicular histotype of thyroid neoplasm was shown in single study (Colamaio et al., 2012). Level of let-7 in circulation of FTC patients seems to be not yet evaluated. Our study can be a first report about increased level of let-7 in plasma of FTC patient, especially in TPO(+) population of EVs. This point was now reflected in discussion section.

2. Can this method (MPO+ EV) theoretically might be applied for any thyroid cancer including more common PTC?

We believe, yes. Since TPO(+) is a marker of follicular thyroid cells, its expression can be expected on surface of vesicles derived by cells of any type of differentiated thyroid cancer including PTC. We would probably not expect to detect this marker on the surface of EVs derived by anaplastic thyroid cancer cells. However, so far it is only assumptions that need to be tested. We plan to do this in next future as well as expand this approach for others cancer, for instance, isolation and analysis of MeprinA(+) / GAL4(+) EV for colorectal cancer monitoring.

3. Are there any doubts about methodology or any risk of technical bias? If yes, this should be discussed. 

Of course, the technical issues always exist. First of all, portion of thyroid gland / thyroid cancer -derived vesicles is supposed to count for only minor fraction of plasma EV. So, efficacy of isolation, or at least enrichment, of thyroid-derived EV is a crucial aspect. Second important issue is mentioned in your next comment: TPO is not absolutely tissue-specific marker. Third important issue is possible losing of tissue-specific marker by cell and, consequently, by cell-derived EV during progression and decrease of differentiation of thyroid cancer. In our opinion, all these issues can be addressed by exploring technology with multiple antibodies to different tissue specific markers attached to immune-beads (TPO, TSH-Rec, and others).

Stability and efficacy of antibodies binding to beads present addition (rather technical) issue. Binding via streptavidin and biotin is well established method, however streptavidin is not very stable and labelling of  antibodies with biotin is quite expensive and tricky procedure. After experimental validation of tissue-specific EV markers, we plan to explore aptamers instead of antibodies and to switch to so-called “click chemistry” approach instead of streptavidin-biotin. We hope to show results of these studies soon J).

Some of these considerations are added in discussion section as suggested.

III. Major problems:

1. TPO is NOT exclusively expressed only in thyroid cancer, but also in the breast, and this issue needs to be carefully discussed (read below).  Godlewska M, et al. Localization of key amino acid residues in the dominant conformational epitopes on thyroid peroxidase recognized by mouse monoclonal antibodies. Autoimmunity. 2012;45(6):476–84.  

Yes. This issued is now discussed in the text.

2. Some question about Quantitative Real-Time RT-PCR (qPCR) a) How the probes were designed or they were applied previously? If they designed how those were tested? They should be if this not commercial products.

All systems for RT-qPCR were designed by traditional approach applying stem-looped RT-primers. Probes were designed to be complement to non-miR-specific part of the cDNA and with incorporation of several LNA monomers to increase melting T and to increase specificity. Before use, each system was tested with synthetic miRNA (mimic) serial dilution. These and similar systems for RT-qPCR analysis of miRNA were used in our previous studies ((Samsonov et al., 2016), (Ivanov et al., 2018), (Titov et al., 2019). To save the space, we did not include data of RT-qPCR system validation however they can be provided by request.

3. Reference Ct of miR-191 – why this one? especially in the context that previously it has been reported that it might be down-regulated in thyroid? This needs to be explained, see the article below. M Colamaio et al. 2011 miR-191 down-regulation plays a role in thyroid follicular tumors through CDK6 targeting.

Mir-191 was selected because it is considered as possible reference for plasma circulating miRNA (recommended by “miRCURY LNA miRNA Focus PCR Panels” manual from Exiqon and proposed by some reports (Danese et al., 2017). Several other molecules, like snRNA-U6 and hsa-miR-151a-3p,-197-3p, -99a-5p and-214-3p validated for thyroid tissue previously (Titov et al., 2016) were included in this study while were not stably expressed in plasma EV. Taking in account a low fidelity of normalization to miR-191, more robust method of normalization to global Ct mean was used in this study as well.  

4. The statistical analysis details are missing. Sample distribution P/NP, tests applied, furthermore correlations: which test was applied Spearman or Pearson’s?

Distribution of the analyzed parameters cannot ne estimated as parametric. In all cases, when two groups of samples were compared, statistic significance of observed difference was evaluated by non-parametric Mann-Whitney test. When three and more groups of samples were compared, non-parametric Kruskal Wallis test was used. Corresponding paragraph was added to Material & Methods section.

5. In my opinion, the structure of the article is very unusual:

a. The introduction has some parts with results description (line 76-78)

            Removed

b. Methods were mix with results in example in lines 260-266

Procedure of dot-blotting and used antibodies are described in Material & Method section. Details of dot spotting are mentioned in Result section focused on TPO(+)EV concentration during isolation workflow. It was done to better explain observed results.   

c. Why the figures and tables are an additional subsection of results section “3.6. Figures, Tables and Schemes”?

Because it was supposed by template downloaded from MDPI. In printed version of article, Figures are supposed to appear near the text where they are mentioned.

d. The results are discussed in the results section (205-208, 270-275 lines) what shorts the discussion to just part of the page and includes just three citations 21-23?!?

Explanation of links between consequent experiments in Results section may resemble discussion, however it may help to understand reason of each next step of study. Discussion section is now extended by additional paragraphs.

III. Minor problems

1. The table with Oligonucleotides used for the RT-qPCR analysis of miRNA might be easily added to supplementary materials? 

Yes.

2 As a potential reader, I really like the study design flow chart and some final remarks or conclusions, however I leaving this, up to the author's decision. 

We included Figure 1 with work flow of study and expanded Discussion section.

3. In figure 9 Receiver operating characteristic (ROC) curves present AUC I guess? (would be beneficial to see exact sensitivity and specifity)

Yes, it’s AUC. We guess, that the main purpose of this study was to demonstrate than TPO(+)EV can be isolated (1) and analysis of miRNA from TPO(+) fraction of plasma EV has higher diagnostic potency then analysis of miRNA from total population of plasma EV. This was shown by ROC analysis. We developed and tested a new diagnostic approach but method is not ready for final evaluation. In order to increase diagnostic power of this approach, the list of tested miRNA should be increased and algorithm of multiple miRNAs evaluation should be developed. Then it becomes necessary to estimate sensitivity and specificity of new diagnostic method.

4. In the discussion, there are some research articles in my opinion that are missing suggesting the potential application of liquid biopsy in thyroid cancer diagnostics since TPO EV might also be used in other types of thyroid cancer?

Discussion section was expanded.

5. The table with patients' characteristics would be a good idea. Moreover, inclusion/ exclusion criteria are needed.

Requested Table is included as well as inclusion/ exclusion criteria are added into the text.

6. First of all, I don't feel qualified to judge about the English language and style. However, in my opinion some mistakes of the manuscript I have noticed should be corrected to improve the quality of the text:

All mistakes are corrected.

Round 2

Reviewer 2 Report

Dear Sirs

The authors significantly improved the manuscript and included also optional corrections/suggestions. Discussion has been fundamentally changed and increased by over a dozen publications. Moreover, the article structure has been improved. The idea is very interesting and should be appreciated.

However, please recheck carefully whole text especially new parts, below few mistakes I have noticed:

Line 173 : were treated instead of where treated?

Line 274: aldehyde instead of aldheyde-sulfate?

Line 994: what means (2)?…” for EV isolation (2) and”

Overall Recommendation: The paper is in principle accepted after slight/minor revision (editor for evaluation).

Author Response

Thank you for the remarks!

Indicated mistakes are corrected.

With best regards,

Anastasia Malek